# Determinants of Clinical Decision Making under Uncertainty in Dentistry: A Scoping Review

**DOI:** 10.3390/diagnostics13061076

**Published:** 2023-03-13

**Authors:** Alexander Ivon King Murdoch, Jordan Blum, Jie Chen, Dean Baziotis-Kalfas, Angelie Dao, Kevin Bai, Marina Bekheet, Nimret Atwal, Sarah Sung Hee Cho, Mahen Ganhewa, Nicola Cirillo

**Affiliations:** 1Melbourne Dental School, The University of Melbourne, Parkville, VIC 3052, Australia; 2CoTreat Pty Ltd., Melbourne, VIC 3000, Australia; 3School of Dentistry, University of Jordan, Amman 11942, Jordan

**Keywords:** medical errors, decision making, diagnosis, clinical reasoning, therapeutics, heuristics, bias, dentistry, patient preference, education, informatics, artificial intelligence, big data, guideline

## Abstract

Clinical decision-making for diagnosing and treating oral and dental diseases consolidates multiple sources of complex information, yet individual clinical judgements are often made intuitively on limited heuristics to simplify decision making, which may lead to errors harmful to patients. This study aimed at systematically evaluating dental practitioners’ clinical decision-making processes during diagnosis and treatment planning under uncertainty. A scoping review was chosen as the optimal study design due to the heterogeneity and complexity of the topic. Key terms and a search strategy were defined, and the articles published in the repository of the National Library of Medicine (MEDLINE/PubMed) were searched, selected, and analysed in accordance with PRISMA-ScR guidelines. Of the 478 studies returned, 64 relevant articles were included in the qualitative synthesis. Studies that were included were based in 27 countries, with the majority from the UK and USA. Articles were dated from 1991 to 2022, with all being observational studies except four, which were experimental studies. Six major recurring themes were identified: clinical factors, clinical experience, patient preferences and perceptions, heuristics and biases, artificial intelligence and informatics, and existing guidelines. These results suggest that inconsistency in treatment recommendations is a real possibility and despite great advancements in dental science, evidence-based practice is but one of a multitude of complex determinants driving clinical decision making in dentistry. In conclusion, clinical decisions, particularly those made individually by a dental practitioner, are potentially prone to sub-optimal treatment and poorer patient outcomes.

## 1. Introduction

Central to medical ethics is the premise “primum non nocere”—first do no harm. The dental profession has taken significant strides in the past century to reduce patient harm. However, the quality of provision of dentistry globally remains highly variable, and this is often partially attributable to clinician and patient factors. For example, a recent study found that 21.1% of dentists reported wrong tooth extraction [1]. Clearly, diagnostic and treatment planning errors contribute to patient harm. In a primary setting, dentists routinely diagnose and plan treatment in isolation, relying largely on their own decision-making capabilities. Clinical decision-making utilises both cognitive (analytical) and intuitive (non-analytical) processes to consolidate the complex information available into a treatment plan [2,3,4]. Decisions are made under many unpredictable conditions, including patient specific factors or preferences, financial constraints, technical factors regarding treatment, and the clinician’s knowledge, experience, emotional state, and fatigue [5,6].

Individual judgements are often made intuitively on a limited number of heuristics and biases. This process streamlines decision making into a simpler task, which enables the decision to be made in an efficient and timely manner. Fast thinking, though, can lead to severe and systematic errors [7]. Clinicians relying on the intuition-based heuristics such as ‘representativeness’ and ‘availability’ may disregard probabilistic, systematic, and check-listed thinking [8].

In an age of evidence-based dentistry, algorithms, informatics, and big data, there is real hope for further reductions to patient harm by enhancing clinician decisions. Contemporary software has already demonstrated its utility in assisting the diagnosis and management of dental diseases across a variety of specialties. Foreseeably, rapid improvements in AI technologies will likely make such clinical support ever more valuable and common place [9,10]. 

Clinical decision-making guidelines and augmentations to patient informed consent both have parts to play in limiting the role of clinician cognitive biases and heuristics in patient harm [11,12,13,14].

Decision making under uncertainty has now been researched for over 60 years and has been widely applied in economic and geo-political situations. There is, however, a paucity in its application to health care—particularly dentistry. Given that dental treatment plans can also be viewed as clinical decision making under uncertainty, the purpose of this scoping review is to investigate the underlying process of decision making in clinical dentistry and the factors that influence an individual clinician when diagnosing and treatment planning a case.

## 2. Methods

### 2.1. Search Strategy

A preliminary search strategy was devised initially in MEDLINE Ovid. During the preliminary search, keywords relevant to the topic ‘decision making under uncertainty within dentistry’ were included to gauge the range of literature available. These included search terms: ‘group OR collective’, ‘diagnos*’ OR ‘deci*’, ‘dent*’. The search string was then further developed, and sensitivity analysis of several searches was undertaken using some landmark papers as guiding references.

The search strategy was created based on the three keywords associated with the review topic. 

‘dentist* OR dental’ restricted the records to the oral healthcare field.‘decision OR diagnosis OR treatment OR management OR guideline’ limited result articles to diagnosis and treatment planning, which are both vital clinical decisions in dentistry.‘decision making process OR heuristi* OR intuition OR clinical reasoning OR clinical judgment OR collective intelligence OR informatives’ narrowed down papers to involve the thought processes of clinicians involved in the studies.

These three domains were combined with Boolean operators as appropriate. The final search string was conducted through PubMed as follows: (dentist*[Title/Abstract] or dental[Title/Abstract]) AND (decision making process[Title/Abstract] or heuristi*[Title/Abstract] or intuition[Title/Abstract] or clinical reasoning[Title/Abstract] or clinical judgement[Title/Abstract] or collective intelligence[Title/Abstract] or informatics[Title/Abstract]) AND (decision[Title/Abstract] OR diagnosis[Title/Abstract] or treatment [Title/Abstract] or management[Title/Abstract] or guideline[Title/Abstract]).

### 2.2. Eligibility Criteria

Articles were evaluated initially based on their titles and abstracts by 2 researchers independently. If a disagreement occurred, a third researcher served as a judge to make the final decision. Adhering to the Cochrane guidelines, a lenient policy was adopted during the selection process where there were any uncertainties. In cases of further disagreement or ambiguity, a meeting was held to discuss the decision of inclusion, with the senior authors (MG and NC) to determine the final outcome. 

Articles were included in the scoping review if they satisfied the following criteria:Primary healthcare setting, focusing on the dental field.Studies conducted or available in English.Making clinical decisions in diagnosis, treatment, or patient management.Clinical decisions can be made on real or hypothetical cases.

Articles were excluded from the scoping review due to any of the following criteria:The contents of the article were irrelevant to the scoping review and did not discuss clinical decision making within the dental field.Review articles that lacked original research results.Article was not available in English.Article full text was not available.

### 2.3. Data Extraction and Analysis

The full texts of the articles included were accessed and data extraction was undertaken using a master table. Additional articles that did not meet the inclusion criteria were excluded if missed by the initial screening. The information extracted to the master table included study design, number of participants, data collection method, and main conclusion drawn.

## 3. Results

As of 4 June 2022, our search strategy on PubMed yielded 478 results. Each article was assessed for eligibility and 64 were included in the qualitative synthesis (Appendix A). Of the 81 excluded (based on exclusion criteria), 45 lacked original research, 29 were not relevant, 6 were inaccessible, and 1 was excluded as the participants were not from a dental background. Details of the article selection process are illustrated by a PRISMA flowchart (Figure 1).

### 3.1. Study Characteristics

By year of their publication, studies included ranged from 1991 to 2022. Of the 64 articles, 46 were published in the last 10 years (2012 to 2022), greatly exceeding the number of articles that were written before from 1991 to 2011.

Studies were based in 27 countries, but were predominantly from the United States (*n* = 13) and the United Kingdom (*n* = 13). Out of all the studies except 1, which represents ‘qualified academicians’, 12 involved participants in a dental setting either as patients, dental students, general dentists, dental specialists, dental hygienists, or assistants. Dental specialties were diversely represented, with orthodontists, paediatric dentists, endodontists, prosthodontists, oral maxillofacial surgeons, oral medicine specialists, and dento-maxillofacial radiologists.

All studies were observational studies except four [15,16,17,18], which were experimental studies. The types of observational studies included were a mixture of descriptive studies (case report, case series, cross-sectional, qualitative studies) and analytic studies (cohort, case-control). The most common methods of data collection were questionnaires/surveys and semi-structured interviews. Only one randomised controlled trial was reviewed [17], and the Delphi method was used in four studies [14,19,20,21].

Under the umbrella of decision making under uncertainty in dentistry, the roles of six major recurring themes were identified: heuristics and biases, clinical factors, clinical experience, patient preferences and perceptions, artificial intelligence and informatics, and existing guidelines.

Of note, the first article about artificial intelligence & informatics was published in 2008. However, most studies on this topic are concentrated between 2017 and 2022 (7 of 10 studies), suggesting that it is a rising field of research. Furthermore, 10 of the 11 studies investigating dental patient perception, preference, and patient-centred care were published before 2016. 

The main findings of the six themes are reported in the following paragraphs. 

### 3.2. Heuristics and Biases Related to the Decision-Making Process

Of the 64 studies included in the review, 13 examined the effect of various common heuristics and biases on the decision-making process in dental settings (Table 1). 

Decision making was demonstrated to be impacted due to clinician education and preferences as well as patient characteristics. Practitioners were more likely to pursue procedures that they enjoyed and were influenced by their “frame of mind” [26]. In high-cost healthcare interventions, e.g., implants, clinical decisions made by practitioners are influenced by assumptions about patient characteristics and financial status [29,32]. Decision making was influenced by dentists’ self-confidence [32], intuitive shortcuts based on prior experiences [33], and differential understanding of the progression of disease [25].

Biases that affect clinical decision-making include those that affect the dentists’ perceptions of the patient. Many patients with a history of addiction and mental illness revealed they felt excluded from the decision-making process [28]. Social and behavioural determinants were also demonstrated to impact decision making [22], suggesting that both implicit and confirmation bias alter this process. Age was a major theme in two studies [27,30] discussing endodontic treatment. Older patients were more likely to receive more invasive treatment in asymptomatic and symptomatic cases, despite no firm evidence supporting age significantly altering the effectiveness of vital pulp therapy [27]. 

### 3.3. Clinical Factors Affecting Clinical Decision-Making 

Study results from 15 included papers revealed that clinicians rely on subjective clinical interpretation of investigations such as patient history, visual oral examinations, and radiographic interpretations to form a diagnosis and treatment plan (Table 2).

Seven studies assessed the influence of patient factors on confidence in clinical decision making [23,37,38,40,43,44,45,46]. The following inherent factors were considered important in clinical decision making: patient age, dental history, clarity of communication, social circumstances, functional aesthetic demands, smoking and alcohol consumption, and medical history [23]. Studies also show that the complexity and severity of the patient’s condition affected diagnostic accuracy, confidence in treatment planning, and the decision to refer [37,38,40,43,44,45,46]. For example, clinicians more confidently selected patients with fewer missing teeth and no bone loss for treatment by implant replacement [40]. Similarly, crowding severity and soft tissue profile influence orthodontic decision making [38]. Referral of potentially malignant diseases to oral medicine specialists depended on lesion colour, cause, location, duration, pattern, and size [43]. The decision to extract periodontally affected teeth or refer to a specialist is influenced by case complexity, dictated by factors such as mobility, severe attachment loss, and radiographic bone loss [45]. In paediatric patients, crying was an accurate indicator of dental anxiety, which influenced the clinical decision to use sedation [46].

Four studies found that the type of treatment had effects on confidence in clinical decision making [23,24,36,41]. Procedural predictability, technical difficulty and risk of iatrogenic damage influenced the clinician’s confidence in decision making [36]. McGeown et al. adds that equipment limits and time and facilities also played a role, while Fu et al. considered timing of clinical findings as important for confidence in treatment planning, where late detection of tooth eruption disturbances can lead to greater complexity [23]. Ilgunas et al. emphasizes the negative effect that time pressure has on accurate clinical decision making in relation to TMD management [24].

Four studies assessed the influence of radiographs on confidence in clinical decision making [35,39,42,47]. Information obtained from radiographs on the depth of a lesion influences the decision to treat. Clinicians were more confident with their decision when lesions involved the inner half of dentine [39]. Despite the importance of bitewing radiographs in the decision-making process, visual examination of caries via ICDAS lead to more accurate diagnosis [42]. Furthermore, clinicians were more confident with their clinical decisions when based off a cone beam CT compared with a periapical radiograph or OPG [35,47].

### 3.4. Clinical Experience Affecting Confidence in Decision Making

There were 13 extracted studies which focused on clinician factors associated with clinical decision making (Table 3). It was revealed that practitioners with more years of experience were associated with greater diagnostic accuracy in comparison to students [48]. Similar conclusions were drawn regarding extraction in periodontally affected teeth [49,50] and prosthodontic cases [51], linking greater experience with well-founded and conservative treatment decisions. However, more experience also led to greater variability in treatment recommendations [52] and did not alter the perceived difficulty [36]. Greater clinical experience was also shown to increase the practitioners’ confidence in producing an accurate diagnosis in dental hygienists [53,54].

Clinician knowledge and education were also influences in decision making [57]. Dentists have an interest in evidence-based practice principles to improve their personal knowledge, skills, and treatment quality [59]. Most dental practitioners are open to integrating new standards into their practice, although outdated procedures were still prevalent under certain circumstances [58] due to non-clinical factors such as NHS regulations. Likewise, in TMJ disc displacement cases, frontline clinicians expressed high degrees of uncertainty due to lack of knowledge, skills, and experience within the area [56]. Finally, the time of graduation for Australian practitioners was correlated to delays in invasive caries intervention of permanent teeth [55]. 

### 3.5. Patient Preferences and Perceptions in Clinical Decision-Making 

Of the extracted articles, 12 included study designs that investigated the impact of patient preferences on the clinical decision-making process in dentistry. A summary of these studies is shown (Table 4). 

Common findings indicate the preference of patients to be active in the clinical decision-making process [34,61,66]. On the contrary, one study reported that significantly more patients had preferences for information rather than a preference for actual involvement [65]. This result parallels the findings of Johnson et al., which reported a specific decision aid can yield a significant improvement in patient knowledge [17]. 

Patient-centred care is an approach to healthcare whereby care decisions are driven by the individual’s needs, preferences and desired outcomes [67]. Dental literature shows that by using a patient-centred method to generate information, patients will feel more involved in the decision-making process, and a better patient-practitioner relationship can be established [62,63]. No conclusive evidence supports a definitive association between patient preferences and their gender, as some studies reported a significant difference while others did not [64,65]. However, patient anxiety was shown to influence the decision-making process, and its identification was important in assisting dentists to achieve an ultimately successful outcome for their patients [46]. 

Studies reported that shared decision making was based on both clinical and contextual factors such as patients’ views [26,60]. One study, however, reported that shared decision making is more limited in high-cost intervention settings [29]. While a preference for shared decision making proves to be the cornerstone finding of these studies, evidence suggests that both patients and practitioners carry preferences that affect the decision-making process [34]. 

### 3.6. Artificial Intelligence and Informatics in Decision-Making

Out of the included studies, 11 of the papers included in this review were considered as relating to AI (artificial intelligence) and informatics (Table 5). Six of these papers developed and tested an intelligent AI model, of which two investigated the clinical viability of an AI model in making classifications from panoramic radiographs [18,68]. The other four papers [16,19,21,69] evaluated the effectiveness of incorporating AI models in making clinical diagnoses.

This review also includes two survey studies with the intention of ascertaining the receptibility of computer programs in the teaching of dentistry [72,74]. The remaining papers include a case presentation demonstrating the digital orthodontic workflow and its advantages/disadvantages [71], a cohort study assessing the possibility of standardising dentally relevant diagnostic terminology in electronic records [73], and an analysis of information.

### 3.7. Use of Existing Guidelines in Clinical Decision Making

Of the extracted studies, 11 studies outlined guidelines for clinical decision making in dentistry, as seen in Table 6. One study proposed use of a framework for managing data elements in oral disease diagnoses, which achieved nearly 95% agreement amongst experts, enabling consistency and transparency [19]. 

While it has been suggested that clinical guidelines often oversimplify treatment decision-making regarding implants, numerous studies highlighted the importance of considering multiple factors before selecting, preparing, or saving implants, particularly the patient’s medical history, bone quality, and implant location [30,75,76,77]. This was echoed by studies investigating surgical, extractive, and sedative decision making, which suggest consideration of patient factors such as medical history, including oncological prognosis and the level of dental anxiety [78,79,80,81].

Additionally, the use of assessment models for more systematic, transparent, and reproducible decision-making, such as for instrument selection and caries assessment, has been explored. While some assessment models had potential to be clinically applicable, others only found agreement with clinical judgement sometimes [14,82].

**Table 6 diagnostics-13-01076-t006:** Extracted articles that investigated the use of existing guidelines in clinical decision making under uncertainty.

Author	Year	Main Findings	Study Design
Abbas et al. [81]	2022	An indication of a sedation requirement tool can be used in patients with dental anxiety and those needing complex planned dental treatment.	Cross-sectional
Deniz et al. [14]	2022	Proposed a framework of multi-criteria decision making for Nickel Titanium instruments selection.	Case study
Amadi et al. [80]	2021	Treatment for geriatric populations must consider patient factors, such as bone atrophy, reduced tissue healing capacity, and medical history.	Case report
Korsch et al. [75]	2021	Comorbidities may lead to general refusal of pre-implantological methods to treat atrophic tooth gaps.	Clinical trial
Tarnow et al. [77]	2021	Key factors influencing decision making regarding implants in malposition include their restorative position, disease status, and depth.	Case report
Ehtesham et al. [19]	2020	A consensus-based framework for essential data elements in differential diagnoses of oral diseases was successfully made.	Cross-sectional
Eliyas et al. [78]	2020	Head and neck cancer patients on long-term antiangiogenic medication are at higher risk of complications from dental extractions.	Case report
Brescia et al. [79]	2019	One must consider many factors to select the appropriate surgical approach for removal of foreign modies in the maxillary sinus.	Case series
Chatzopoulos et al. [76]	2018	Root canal treatment exhibited a higher failure rate than implant treatment. Selection of either should be based on multiple factors, such as age and anxiety.	Cross-sectional
Hänsel Petersson et al. [82]	2016	Caries risk assessment based on clinical judgement and the Cariogram model gave similar results for patients predicted at a low level of future disease.	Cross-sectional
Su et al. [30]	2014	There is no irrefutable clinical guideline that can be followed for whether a tooth should be replaced with an implant, or be treated and maintained.	Cross-sectional

## 4. Discussion

Decision making in dentistry is complex and dependent on a number of factors. In this scoping review, we identified the following: heuristics and biases, clinical factors, clinical experience, patient preference, intelligence and informatics, and existing guidelines. These factors can substantially influence the clinical decisions made in regards to diagnosis and treatment planning when conducted by a single dental practitioner, potentially leading to suboptimal treatment and poorer patient outcomes.

### 4.1. Heuristics and Biases

Rather than an objective analysis of a case, post hoc justifications of courses of care are made primarily due to intuitive reasoning [23]. Analysis of these “diagnostic thinking processes” revealed that inter-clinician variability was not due to inadequate diagnostic process. Rather, this variability was attributable to the concepts signaling the need to intervene, which are influenced by prior experience. Since dental students are exposed to environments with high levels of decay, they are primed to expect decay and hence place restorations regularly. [33]. Diagnostic processes could incorporate two processes to assist clinical decision making. The first process involves pattern recognition and the second process is explicitly analytical. This allows clinicians to appropriately treat typical presentations and adopt a more analytical perspective in atypical cases [31].

Lack of education about certain disorders, lack of insight in ability, and differential interpretations altered treatment outcomes. Analysis of TMD management by GDPs determined that some patterns for treating TMD were inadequate and reflected a “culture of generalisations”, highlighting that lack of education leads to substandard care [24]. Clinicians’ current state of mind and enjoyment of certain procedures predisposed them to pursue certain treatments [26]. “Weaker students” often lacked insight of their true competence and overestimated their abilities, which may result in poorer outcomes. Men more often overestimate their abilities than women [32]. Another cause of the high individual variation for retreatment is due to the clinicians’ interpretation of where on the continuum of periapical health a patient is. This results in disagreement as to what constitutes optimal treatment [25]. Treatment provided is directly affected by clinician factors, such as clinician education, frame of mind, and assumption of a patient’s financial status [26,29,32]. Therefore, it may be postulated that collective diagnosis may reduce sources of bias and error when treatment planning. Whilst the effects of collective intelligence on decision-making accuracy in medicine is relatively well studied, our results reveal a scarcity of literature on its potential application in dentistry [83]. 

### 4.2. Clinical Factors 

A number of clinical factors play a role in the practitioner’s confidence when making diagnostic and treatment decisions. These can be broken down into three groups: patient factors, treatment complexity, and clinical information obtained from examination and further investigations. The magnitude of the effects that these factors have on decision making is dependent on the particular case. For example, smoking status plays an important role in diagnosis of potentially malignant lesions in the mouth [43], but not so much when treatment planning for composite restorations. Results also identified some non-clinical factors that included time, facilities, and dental anxiety, which further affected the clinician’s decision-making ability.

### 4.3. Clinical Experience

The experience of the clinician is correlated with confidence in providing accurate diagnoses and appropriate treatment plans. This is also seen in psychology [84], where experience was a strong predictor for diagnostic accuracy, primarily due to exposure to more clinical cases. The education of the clinician influences core principles and philosophies behind treatment decisions, with more recent education and professional development producing clinicians that tend to practice with updated and conservative philosophies, preserving tooth structure where possible.

The intermediate phenomena observed in a specific study [48] suggests that clinical experience allows for an ‘encapsulated’ mode of clinician information processing and diagnosis compared to a more ‘elaborated’ response from dental students, possibly due to factors such as appointment time constraints and profit-driven pressures that are applied to practicing dentists. Practitioners may also become more pragmatic in their clinical practice and decision making [55], although more research is still needed within the area to investigate the reasoning behind these changes.

### 4.4. Patient Preference 

Ultimately, the choice of action to be taken lies in the hands of the patient, although the different values and preferences of patients determine which decision will be taken. The majority of the papers examined indicated a preference for patients to be involved in the decision-making process. Results indicated that while clinical factors are important in this process, they are ultimately overruled by patient preferences [26]. This is likely to have implications in the clinical setting, as the nature of preferences are subject to bias and predispositions. In particular, both anxiety and patients’ impressions of dentists’ personality are shown to mediate both the treatment and decision-making process [34,46]. Overall, a desire for shared decision making proves common to the majority of studies that examined the effects of patient preference, although gaps in the literature do exist. Most of the information published that addresses this factor are a part of specific dental-related cases reports. As such, prospective studies may wish to design a study that focuses on the variety of patient preferences and perceptions that impact the treatment and decision-making process in dentistry. This will likely assist in accurately dissecting the decision-making process, thus allowing for the possibility of improved treatment outcomes and concord in the shared decision-making process.

### 4.5. Intelligence and Informatics 

Choi et al. and Tuzoff et al. highlight the possibility of AI usage to aid in interpretation of dental radiographs [18,68]. Several papers—in addition to those included in this review—have advocated for the use of computer-aided diagnosis (CAD) as a means of providing an interface for managing and simplifying complex variables involved in the traditional clinical diagnosis process [19,21,69,74,85]. The use of computer programs in dentistry has been rapidly evolving for quite some time now with the advent and popularisation of digital workflow systems providing a way to improve communication—both between practitioners and with patients—as well as a means of improving overall efficiency and decreasing laboratory workload [70,71,86,87]. 

The papers in this review all explored the efficacy of AI with a positive outlook. The shortcomings of these programs, however, cannot be ignored. The studies included were primarily cohort or cross-sectional studies with few participants, rather than clinical trials, and studies that tested the clinical efficacy of an AI program did so using simulation cases as opposed to real-life clinical cases. It is evident that further clinical testing of these programs are required; however, a fundamental constraint persists in measuring the accuracy of clinical decision making due to its subjective nature and its inherent biases, which leads to inter-practitioner discrepancies. Thus, obtaining an objective measure of clinical decision-making accuracy may be considered almost impracticable. Nevertheless, it is our optimistic perspective that AI-based technology will persist in expanding its horizons and potentially assimilate into clinical dentistry in the future.

### 4.6. Use of Existing Guidelines 

Clinical practice guidelines include practice recommendations informed by unbiased systematic reviews of evidence, and an assessment of the benefit and the harms of alternative treatment options. As such, clinical guidelines guide the clinical decision making of dental practitioners by standardising appropriate healthcare for specific clinical circumstances. For example, the framework suggested by Ehtesham et al. of essential data elements to form a differential diagnosis for oral diseases achieved a high level of agreement, and has the potential to become a uniform professional criterion [19]. However, the study may have been limited by geographic bias, and future studies would therefore benefit from wider participant sampling. 

Dentists should consider the specifics of the individual patient, and assess the applicability of existing guidelines. The guidelines suggested by numerous studies emphasised patient-centred care and the importance of considering patient factors during treatment [75,76,78,81]. For example, a history of endocarditis and bisphosphonates may lead to a general refusal of implants [75,76]. However, these studies had low response rates, a limited number of patients, and were retrospective, which may limit their ability to generalise results. Future studies should thereby aim to investigate a greater number of patients and be prospective studies.

## 5. Conclusions

Decision making in dentistry, despite its importance to patient care, remains an under-researched domain. Clinical judgement is highly complex and dependent on experience; discrepancies between practitioners is ultimately inevitable, with cognitive heuristics and biases being formed from personal education, clinical experience, and self-confidence. Unsurprisingly, a clinician’s inclination towards procedures, estimation of their abilities, and psychological state can impact the quality of care a patient receives. Ultimately, shared decisions are made between the practitioner and patient with preferences and contextual factors influencing the agreed treatment plan.

Although novel, AI models and informatics have the potential to assist traditional clinical decision-making, but larger and more rigorous studies are warranted to determine if there is significant clinical benefit to be obtained. Reference to guidelines may update and supplement personal experience. Realistically, individual diagnostic or treatment planning errors are unlikely to be fully eliminated due to the various human elements that define clinical dentistry.

There are inherent limitations to scoping reviews, including the potential to miss articles due to the database selection. “Gray” literature, or literature of borderline relevance, may have been excluded, and studies not published in English were not reported here. The results of our study reveal a network of determinants influencing a dentist acting as a sole decision-maker under uncertainty. Inconsistency in treatment recommendations between clinicians is a real possibility, increasing the risk of sub-optimal treatment being provided and poorer patient outcomes. Evidently, there exists a need for additional research into methods of improving clinical decision-making accuracy in the dental setting, which aim to minimise error and place patient safety at the forefront.

## Figures and Tables

**Figure 1 diagnostics-13-01076-f001:**
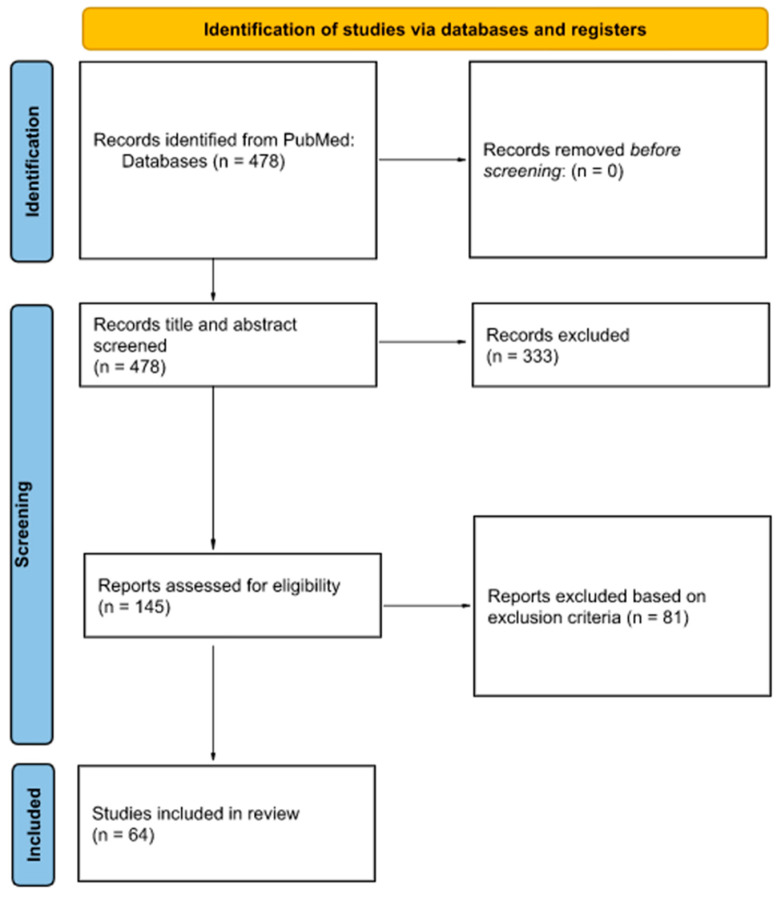
Flowchart for study selection in accordance with Preferred Reporting Items for Systematic reviews and Meta-Analyses extension for Scoping Reviews (PRISMA-ScR) guidelines.

**Table 1 diagnostics-13-01076-t001:** Extracted articles that investigated heuristics and biases related to the decision-making process.

Author	Year	Main Findings	Study Design
Meyer et al. [22]	2022	Caries treatment decisions varied widely among the 10 paediatric dentists. More effort should be placed on calibrating decision-making processes for caries in the primary dentition.	Qualitative, cross sectional
McGeown et al. [23]	2022	Dentists were more likely to extract teeth under general anaesthesia in patients with disabilities.	Qualitative, cross sectional
Ilgunas et al. [24]	2021	Dentists ‘combined own competence and other’s expectation in the desire to do the right thing’. TMD decision making process presented challenges and complexity.	Qualitative, inductive
Helayl Al Waqdani et al. [25]	2021	R4 residents were more likely (not significantly) to choose retreatment or follow up compared to R3 residents.	Retrospective
Dawson et al. [26]	2021	In coronal restorations of a root-filled tooth, the GDP’s decision-making process was based not only on clinical factors, but also on decisive context––factors and consideration of the patient’s views.	Qualitative
Careddu et al. [27]	2021	Patient symptoms and age significantly influenced the decision-making process and invasiveness of endodontic treatment. Young AIE members preferred calcium hydroxide, while older clinicians preferred hydraulic calcium silicate.	Structured online questionnaire
Brondani et al. [28]	2017	Patients with mental illness and addiction perceived being treated differently by practitioners as they felt they were not involved in the decision-making process and felt they were treated as “different” and “unworthy”.	Cross sectional
Vernazza et al. [29]	2015	When making decisions about a high-cost dental intervention where the patient meets the costs directly, shared decision making is limited.	Qualitative
Su et al. [30]	2014	Clinician training history is important in addition to traditional patient/tooth-associated factors.	Cross sectional
Khatami et al. [31]	2012	Clinical reasoning is a non-linear process. Participants had different strategies and would often go back and forth to determine diagnoses and treatment plan.	Cross sectional
Devlin [32]	2012	Self-assessed competence, skill and judgement affects the prognosis of single surface amalgam restorations.	Cross sectional
Maupomé et al. [33]	2000	Heuristics detected link features common to the GM model and to an indirect pattern-recognition model, whereby reliance on visual/tactile concepts facilitates the acquisition of a clinically meaningful image.	Qualitative
Redford et al. [34]	1997	Patients and dentists bring biases which affect the treatment decision-making processes. Dentists use intuition and judgement to depart from ideal and or modify treatment plans on a patient-to-patient basis. Patient’s impressions of dentists’ examination style, personalities, and ability to relate to them as individuals seem to mediate both treatment acceptance and willingness to participate in the decision-making process.	Cross Sectional

**Table 2 diagnostics-13-01076-t002:** Extracted articles that investigated clinical factors affecting clinical decision making under uncertainty.

Author	Year	Main Findings	Study Design
McGeown et al. [23]	2022	Systemic and analytic decision making is based on a number of clinical factors, which, along with heuristics, biases, and patient preference, influenced the clinical decision-making process.	Qualitative, cross sectional
Luz et al. [35]	2022	Clinical confidence in treatment decisions improved with CBCT scans compared to PA radiographs.	Prospective cohort study
Liew et al [36].	2021	Clinicians were less confident with their clinical decisions when they pertained to a difficult case.	Qualitative, cross sectional
Ilgunas et al. [24]	2021	There is a lack of confidence in decision making for management of TMD due to a number of patient, clinician, and organisational factors.	Qualitative, inductive study
Kafantaris et al. [37]	2020	Along with interpretation of specialist clinicians, decisions for treatment planning congenitally missing teeth were dependent on case characteristics and patient factors such as age.	Retrospective cohort study
Evrard et al. [38]	2019	Clinical decision making was most strongly influenced by clinical factors, namely, soft tissue profile and crowding.	Qualitative, cross sectional
Leal et al. [39]	2019	Confidence in treatment planning for carious primary dentition increased with depth of cavity.	Cross-Sectional
Cosyn et al. [40]	2012	Along with clinical experience, factors such as disease severity and complexity influenced the degree of variability in clinicians’ treatment recommendations for periodontal disease.	Qualitative, cross sectional
Fu et al. [41]	2012	Timing of clinical findings played an important role in clinician’s decision-making.	Case report
Diniz et al [42].	2011	ICDAS scores were more diagnostically accurate when compared with bitewing radiographs alone, leading to better decision making.	Prospective cohort study
Brocklehurst et al. [43]	2010	Correct referrals of malignant disorders relied on patient factors such as age, smoking status, and alcohol consumption, as well as lesion factors such as location, colour, and size.	Qualitative, cross sectional
Korduner et al. [44]	2010	The decision-making process regarding shortened dental arch treatment planning is based on a number of patient-related items	Qualitative, cross sectional
Moireira et al. [45]	2007	Clinical factors such as mobility, severe attachment loss, and radiographic bone loss influenced clinicians’ confidence in treatment planning and referral decisions	Qualitative, cross sectional
Holmes et al. [46]	2005	The decision to sedate pediatric patients with dental anxiety is based on the operators’ accurate identification of anxious patients.	Qualitative, cross sectional
Danforth et al. [47]	2003	CBCT proved to be a more effective tool compared to standard film radiography in decision making for impacted wisdom teeth.	Case report

**Table 3 diagnostics-13-01076-t003:** Extracted articles that investigated clinical experience affecting confidence in decision making under uncertainty.

Author	Year	Main Findings	Study Design
Mecler et al. [49]	2022	Practicing dentists were less conservative when providing treatment decisions to both cases, with students opting to maintain the teeth despite the indication for extraction. Specialists in implant dentistry and periodontics were also more likely to extract the teeth. Clinicians with less than 20 years of experience were also demonstrated to be more inclined to maintain the teeth compared to specialists with more experience.	Qualitative, cross sectional
Liew et al. [36]	2022	Treatment decisions for endondontically involved teeth were mainly influenced by perceived predictability and difficulty of the procedure, risk of tooth damage, and patient preference. Specialty postgraduate training also greatly influenced the treatment decision made.	Qualitative, cross sectional
Swigart et al. [54]	2020	Expertise and confidence in the diagnosis of oral health issues by dental hygienists was reported to be directly linked to clinical experience.	Qualitative, cross sectional
Keys et al. [55]	2019	Australian dental clinicians varied greatly with their treatment decisions for carious lesions. Significant factors included dentist’s age, university of graduation, practicing state, decade of graduation, frequency of treating children, and affected restorative threshold.	Qualitative, cross sectional
Tolentino et al. [50]	2019	Greater years of clinical experience was correlated with more well-founded decisions for extraction of periodontally affected teeth. Periodontists were less likely to extract than general clinicians, and clinicians with more experience were more inclined to extract teeth.	Qualitative, cross sectional
Al-Baghdadi et al. [56]	2019	General dentists had greater uncertainty in diagnosisng and managing temporomandibular disc displacement patients compared to oral maxillofacial surgeons, which is primarily due to lack of knowledge, training, and experience.	Qualitative, cross sectional
Bishti et al. [51]	2018	Prosthodontists with more than 20 years of clinical experience were less likely to prescribe implants as a treatment option than those with less than 20 years of experience.	Qualitative, cross sectional
Korduner et al. [57]	2016	Clinician experience and working with colleagues are shown to significantly influence the treatment decision-making process in prosthodontic cases.	Qualitative, cross sectional
Williams et al. [53]	2014	Clinical training confidence in dental and dental hygiene students was associated with greater odds of appropriate referral for periodontal disease. Most dental students are able to identify critical risk factors that would suggest a periodontal referral, whereas some dentists tend not to refer until severe bone loss has occurred.	Qualitative, cross sectional
Maidment et al. [58]	2010	Dental practitioners are receptive to newer techniques and materials. Outdated techniques such as dentine pin placement may still be performed due to support from regulatory bodies such as the NHS.	Qualitative, cross sectional
Klomp et al. [48]	2009	Higher year dental students, recent graduates, and experienced clinicians showed greater diagnostic accuracy compared to lower year students. Experienced clinicians also showed lower levels of recall following the case.	Qualitative, cross sectional
Yusof et al. [59]	2008	132 out of 198 participating clinicians believed practice based on current literature improves their treatment quality and overall knowledge. However, barriers limit their access to the evidence and they prefer to refer to colleagues.	Qualitative, cross sectional
Cosyn et al. [52]	2007	Experienced dental practitioners showed most variability in their treatment decisions. Training centres of the dentist played a significant role as it shaped their treatment philosophy.	Prospective Cohort

**Table 4 diagnostics-13-01076-t004:** Extracted articles that investigated patient preferences and perceptions in clinical decision making under uncertainty.

Author	Year	Main Findings	Study Design
Dawson et al. [26]	2021	This study found that dentists’ decision-making process was based not only on clinical factors, but also on decisive contextual factors and consideration of the patients’ views.	Qualitative
Barber et al. [60]	2016	This study revealed consensus supporting shared decision making, with no gender differences being reported in the attitudes of dentists towards decision making.	Quantitative
Vernazza et al. [29]	2015	Findings of this study suggest that paternalistic decision-making is still practices and is influenced by assumptions about patient characteristics.	Qualitative
Azarpazhooh et al. [61]	2014	This study found that the majority of patients valued an active or collaborative participation in deciding treatment for a tooth with apical periodontitis. This pattern implied a preference for a patient-centered practice mode that emphasizes patient autonomy in decision making.	Cross-sectional
Ozhayat et al. [62]	2010	This study found that a specific interview method (SEIQoL-DW) could be used as a useful aid in decision making over traditional history taking. This method allowed patients to nominate and prioritize needs, wishes, and problems, thereby generating more useful information than traditional history taking.	Qualitative & Quantitative
Ozhayat et al. [63]	2009	This study found that a specific interview method (SEIQoL-DW) could be used as a useful aid in decision making over traditional history taking. This method allowed patients to nominate and prioritize needs, wishes, and problems, thereby generating more useful information than traditional history taking.	Qualitative & Quantitative
Johnson et al. [17]	2006	This study explored the use of a novel decision aid for use in clinical decision-making in dentistry, which resulted in a statistically significant improvement in patient knowledge of treatment options.	Randomized Control Trial
Gilmore et al. [64]	2006	This study found that dental patients’ willingness to engage in treatment is influenced by the dentist’s clinical recommendation and the importance of oral health to the patient.	Quantitative
Holmes et al. [46]	2005	The findings support the subjective assessment of anxiety in children; however, objective anxiety measures may assist clinicians in identifying specific fears, which may ultimately aid patient management.	Qualitative, cross sectional
Schouten et al. [65]	2004	Results demonstrated that patients have high preferences for information, but their preferences for actual involvement are significantly lower. No differences were found in relation to patient preference for information and participation as a function of gender.	Qualitative
Watted et al. [66]	2000	This clinical report describes a concept of systematic approach to the treatment of Class II deformities, with emphasis on patient input into the decision-making process being critical for a mutually satisfactory result.	Case Report
Redford et al. [34]	1997	Patients’ impressions of dentists’ examination styles, personalities, and ability to relate to them as individuals seem to mediate both treatment acceptance and willingness to participate in the decision-making process.	Cross-sectional

**Table 5 diagnostics-13-01076-t005:** Extracted articles that investigated artificial intelligence and informatics in clinical decision making under uncertainty.

Author	Year	Main Findings	Study Design
Li et al. [70]	2022	Dental practitioners often consulted medical physicians to seek medical clearance for procedures and key patient medical information. The study highlighted the importance of integrated electronic systems to streamline interdisciplinary care.	Cross-sectional
Choi et al. [18]	2022	Created and tested a deep learning algorithm that, when determining the spatial link between M3 and IAN Canal, performed better than professionals. Following clinical testing, the application could have the potential to narrow patient access to CBCT or prepare for surgical extraction.	Cohort study
Ehtesham et al. [19]	2020	A web-based AI system was developed that has potential to assist specialists in making diagnoses of various oral diseases.	Cross-sectional
Perakis & Cocconi [71]	2019	Combining digital technologies and traditional laboratory procedures could be the answer to maintaining a diverse range of options for restorative materials.	Case study
Ehtesham et al. [21]	2019	Almost complete agreement in the framework for structuring the essential data elements in differential diagnosis in oral medicine. A Delphi decision technique was utilised.	Cross-sectional
Tuzoff et al. [68]	2019	A computer-aided diagnostic aid was developed that was able to correctly identify teeth on an OPG with a level of accuracy and precision comparable to a dental practitioner.	Quantitative
Nam et al. [69]	2018	Testing of an artificial intelligence program to assist with TMD diagnosis based on patients’ word usage in presenting complaint and oral aperture measurements.	Cohort study
Thanathornwong [16]	2018	Testing of an application designed to assess the need for orthodontic treatment in permanent dentition patients.	Cohort study
Deshpande et al. [72]	2017	Positive reception of an app designed to help train student prosthodontists in clinical decision making. Results indicated a significant improvement in clinical reasoning abilities following use of the app.	Cohort study
White et al. [73]	2011	The implementation and utilization of standardized diagnostic codes and terminology in an electronic health record were successfully demonstrated.	Retrospective cohort study
Rios Santos et al. [74]	2008	A computer program designed to assist with simplifying clinical decision making with the aid of tree diagrams was well-received by dental students, newly qualified graduates, and experienced dentists.	Cross-sectional

## Data Availability

Full datasets are available upon reasonable request to the corresponding author.

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
