# Peer review of "Determinants of Clinical Decision Making under Uncertainty in Dentistry: A Scoping Review"

_diagnostics, 2023, doi:10.3390/diagnostics13061076_

Round 1
Reviewer 1 Report
This is very well written paper about very complex subject.
In introduction please transform opening sentence: The Primum non nocere. ‘First, doing no harm’ is central to medical ethics. into- Central to medical ethics is premise primum non nocere "First do no harm".
in sentence: But the quality of provision of dentistry globally remains highly variable due to clinician and 30 patient factors. there is missing many more factors that define the quality and availability of dental services. More can be found in: Peres MA, Macpherson L M D, Weyant RJ, et al. Oral diseases: a global public health challenge. Lancet 2019 394: 249–260. or Watt RG, Daly B, Allison P, et al. Ending the neglect of global oral health: time for radical action. Lancet 2019 394: 261–272.
The material and method section is clearly presented, as well as the Results.
In discussion line 280 Analysis of these “diagnostic thinking 280 processes” revealed that inter-clinician variability was not attributable to a lack of diagnostic process, but rather to the concepts signalling the need to intervene" should be more clear. Please explain better.
In the discussion section, the Use of Existing Guidelines is not well explained. Please provide information on how guidelines impact decission making.
Conclusion also need more specification
Author Response
Please see attached response

Reviewer 2 Report
Well written manuscript
Author Response
Thank you for your comment
Reviewer 3 Report
Manuscript of considerable interest for the dental sector, needs a major revision.
Abstract, to better highlight the results obtained.
Keywords, few and not present on MeSH, add more
Introduction: also add all software that improves clinical activity and hard tissue management, such as intact-tooth published on sensors mdpi
Materials and methods: poorly described, expand them
Very confusing results, reorganize them by highlighting the results obtained
Discussion: Add proactive action using AI as goals and it will activate assisted clinics
Conclusions: rephrase them based on the comments
Bibliography: add references required
Author Response
Please see attached comments

Reviewer 4 Report
Thank you for letting me review the article entitled ‘Determinants of Clinical decision making under uncertainty in dentistry: a scoping review’. The aim of the article is to systematically evaluate the clinical decision processes of dentists.
In my opinion, the article is interesting but the results are presented too peremptorily, above all due to the fact that scoping review is not the best method for obtaining maximum evidence. For example lines 20-23 of the abstract are too peremptory. The authors infer that 'adherence to evidence-based practice is not always driving clinical decision making in dentistry' through a survey method that certainly does not represent the best evidence. Data extracted from the articles included in the review and presented in the text are limited to author names, year and study type; the results of the studies included in the review are not even presented in the supplementary table.
The limitations of the study, although significant, are not presented at all. Furthermore, it is not clear from the results why especially the dentist who acts alone is more prone to error.
In paragraph 4.5 the limits of artificial intelligence are not even reported.
In essence, this article seems to have the real objective of discrediting the diagnostic and therapeutic capabilities achieved so far by dental science, which are determined in the relationship between patient and practitioner, in favor of a greater use of automated / IT decision-making processes ...
Round 2
Reviewer 1 Report
The manuscript has been improved by the suggestions and I just want to congratulate the authors.
Reviewer 3 Report
The manuscript has been properly revised according to the comments, it can be published
Reviewer 4 Report
The manuscript has been improved by the suggestions